# Application of Lytic Bacteriophages and Their Enzymes to Reduce Saprophytic Bacteria Isolated from Minimally Processed Plant-Based Food Products—In Vitro Studies

**DOI:** 10.3390/v15010009

**Published:** 2022-12-20

**Authors:** Dziyana Shymialevich, Michał Wójcicki, Artur Wardaszka, Olga Świder, Barbara Sokołowska, Stanisław Błażejak

**Affiliations:** 1Culture Collection of Industrial Microorganisms—Microbiological Resources Center, Department of Microbiology, Prof. Waclaw Dabrowski Institute of Agricultural and Food Biotechnology—State Research Institute, Rakowiecka 36 Street, 02-532 Warsaw, Poland; 2Laboratory of Biotechnology and Molecular Engineering, Department of Microbiology, Prof. Waclaw Dabrowski Institute of Agricultural and Food Biotechnology—State Research Institute, Rakowiecka 36 Street, 02-532 Warsaw, Poland; 3Department of Food Safety and Chemical Analysis, Prof. Waclaw Dabrowski Institute of Agricultural and Food Biotechnology—State Research Institute, Rakowiecka 36 Street, 02-532 Warsaw, Poland; 4Department of Microbiology, Prof. Waclaw Dabrowski Institute of Agricultural and Food Biotechnology—State Research Institute, Rakowiecka 36 Street, 02-532 Warsaw, Poland; 5Department of Biotechnology and Food Microbiology, Institute of Food Sciences, Warsaw University of Life Sciences (WULS–SGGW), Nowoursynowska 166 Street, 02-776 Warsaw, Poland

**Keywords:** bacteriophages (phages), lytic activity, phage enzymes, depolymerases, saprophytic bacteria, food biopreservation

## Abstract

The aim of this study was to isolate phage enzymes and apply them in vitro for eradication of the dominant saprophytic bacteria isolated from minimally processed food. Four bacteriophages—two *Enterobacter*-specific and two *Serratia*-specific, which produce lytic enzymes—were used in this research. Two methods of phage enzyme isolation were tested, namely precipitation with acetone and ultracentrifugation. It was found that the number of virions could be increased almost 100 times due to the extension of the cultivation time (72 h). The amplification of phage particles and lytic proteins was dependent on the time of cultivation. Considering the influence of isolated enzymes on the growth kinetics of bacterial hosts, proteins isolated with acetone after 72-hour phage propagation exhibited the highest inhibitory effect. The reduction of bacteria count was dependent on the concentration of enzymes in the lysates. The obtained results indicate that phages and their lytic enzymes could be used in further research aiming at the improvement of microbiological quality and safety of minimally processed food products.

## 1. Introduction

High microbiological contamination of raw materials and increasing antibiotic resistance are prompting the food industry to search for unconventional, effective methods of minimally processed food (MPF) preservation [1,2,3]. MPF production involves gentle food processing methods and, thus, the level of microbial contaminants in this type of product is higher compared to highly processed food [4]. Minimally processed plant-based foods with the highest levels of microbial contamination include salad mixes, sprouts, lettuces, fresh vegetables and fruits, and unpasteurized juices [5]. The main source of microorganisms in this type of food is the environment, primarily contaminated soil and water [6]. It is estimated that ca. 30% of consumers worldwide suffer from food poisoning caused by bacterial pathogens each year [7]. The statistics indicate that the number of illnesses is increasing year after year and a possible reason for this is the increasing consumer interest in minimally processed foods [8,9]. Reduced quality of MPF products resulting from the growth and development of saprophytic bacteria constitute a significant problem [10,11,12]. Vegetables are a source of nutrients for microorganisms because of their near-neutral pH, high content of vitamins, minerals, fiber [13], and high water activity [14,15]. Plant-based food products have a short shelf life, even at low storage temperatures [16], due to the presence of saprophytic bacteria [12]. The search for methods to protect food has turned to the use of lytic bacteriophages for this purpose [17,18]. Scientific research shows that the use of phages and their enzymes does not affect the sensory properties of food (i.e., taste, appearance, color, and smell) [12,17,19]. Due to their high host specificity, bacteriophages and phage enzymes are becoming promising prospects in the detection and biocontrol of bacterial pathogens in agriculture, food production and processing, and medicine (phage therapy) [20,21,22].

Bacteriophages are known as an abundant form of life on earth and can be found wherever the potential host–bacteria are present [23]. The total number of bacterial viruses is 10^32^ virions, which is 100 times greater than the number of currently characterized bacteria [24,25]. The presence of bacteriophages in the environment is a natural mechanism to ensure bacterial balance [26]. Widespread occurrence is associated with the presence of phages in various environments, such as sewage, soil, and food products [12]. Their presence was confirmed in commercial vaccines and serums, in the mouth and gastrointestinal tract, skin and hair [18,27,28,29]. Specificity for selected bacteria manifests itself in the ability to infect only a particular species, serotype or strain, preventing the destruction of the physiological intestinal microbiota [30]. Phages owe their effectiveness also to auto replication, which eliminates the need for multiple dosages [31].

Bacterial viruses use several mechanisms to replicate their particles, the most common of which are the lytic and lysogenic cycles [32,33]. In the lysogenic cycle, the bacteriophage genome integrates with the bacterial chromosome providing survival of phage in unfavorable environmental conditions [34]. In the lysogenic cycle, virulence and antibiotic-resistant genes may be transferred between the host and the phage, therefore such phages are used neither in phage therapy nor the development of phage biopreparations. No new viral particles are released during lysogeny [32,35,36]. The lysogenic cycle that leads to the synthesis of progeny viruses represents another mechanism of phage particle replication. Viruses rearrange the host’s metabolism to produce new viral particles. During the release of new viruses, phage enzymes take part in disrupting the host cell’s structures (cell lysis) [37,38,39]. The enzymes that degrade the bacterial cell envelope during phage infection and virion release were characterized. These proteins diffuse into the agarose layer, deprive the envelope of bacteria, and grow around as the phage plaque. The result of this process is the formation of a milky translucence at the edge of the plaque (the so-called ‘halo’) [40]. One of the opportunities that is emerging in phage therapy and the concept of ensuring the microbiological safety of food is the use of phage-derived enzymes [18]. Bacteriophage lytic enzymes (mainly holins and lysins) are produced during the replication cycle of the lytic phage [41,42]. Holins are the most diverse group of phage proteins encoded by more than 150 genes [38,43]. They are most often small proteins (60–155 amino acid residues) that accumulate in the cytoplasmic membrane and have no harmful effect on the cell [10,38]. At some point during phage infection, they cause the formation of pores in the bacterial membrane, which increases its permeability [43,44,45]. This allows endolysin from the cytoplasm to enter the periplasm and damage the peptidoglycan [46]. The result of this process is the lysis of bacterial cells from within and the release of virions. Lysins are a group of proteins produced at the final stage of viral particle replication [46,47,48]. These enzymes reduce the mechanical strength and resistance of the cell wall, leading to bacteriolysis and the release of new virions [49,50]. The function of endolysins and basal lysins in most phages is performed by different proteins [41,43]. However, phages in which the function of these enzymes is carried out by a single protein were also identified [51,52,53]. Endolysins, contrary to basic lysins, are not an integral part of the virion but are synthesized in a bacterial cell as a consequence of infection [54,55]. Each phage encodes several lytic enzymes in its genome, some of which are highly specific, while others may be active against different strains and even species [56,57].

Multiple advantages stem from the use of phage enzymes to eliminate bacteria in the food industry. It is considered unlikely that bacteria will develop resistance against phage enzymes because of the interactions between the enzymes and the peptidoglycan of the bacterial wall [58]. It is worth noting that the application of the enzymes constitutes a targeted action and self-replication does not occur (and therefore there is no risk of horizontal gene transfer as in the case of using whole phage particles). The possibility to obtain enzymes from both lytic and lysogenic phages represents another advantage of this approach [59]. The addition of phages and their enzymes to food can improve the microbiological quality of the product without changing its sensory characteristics [4,17]. Moreover, the exclusive use of phage enzymes in food is not as controversial among the public as the application of whole virus particles [18].

Phages and phage enzymes have several limitations as antimicrobial agents in the food industry. Compared to broad-spectrum antibiotics, they possess limited host specificity, which makes the elimination of other non-pathogenic bacteria difficult. The transfer of virulence factors from the bacterial host used for propagation must be carefully considered before using phages as bio-conservatives [60]. This indicates that virulent phages must be thoroughly selected to avoid the transfer of virulence factors [61]. Gram-negative bacteria are resistant to endolysins due to the presence of a bacterial outer membrane that limits the access of endolysins to peptidoglycan [62]. In particular, endotoxins, which are potential contaminants originating from Gram-negative bacteria, should be removed during the phage purification process due to their side effects on human health [63].

Thus, the aim of this study was to obtain the solutions of bacteriophages and their enzymes with enhanced antimicrobial potential and their use in vitro for the eradication of saprophytic bacteria isolated from minimally processed plant-based food products.

## 2. Materials and Methods

### 2.1. Source of Bacterial Host Strains and Bacteriophage Propagation

Saprophytic bacteria and specific depolymerase-producing bacteriophages were obtained from the Culture Collection of Industrial Microorganisms—Microbiological Resources Center at the Department of Microbiology Prof. Waclaw Dabrowski Institute of Agricultural and Food Biotechnology—State Research Institute (IAFB). Bacterial strains were isolated from minimally processed foods that are available on the Polish retail market (lettuce mixture with beets and spinach). Four phage strains were used in the study: *Enterobacter* phage KKP 3262, *Enterobacter* phage KKP 3711, *Serratia* phage KKP 3708, and *Serratia* phage KKP 3709. The *Enterobacter* phage KKP 3262 was characterized by Wójcicki et al. [12]. Assembled and annotated phage genome sequence was submitted to National Center for Biotechnology Information (NCBI) under OK210076 accession number. The other three bacteriophages are novel and will be characterized in our future study. The source of the bacteriophages was municipal sewage (sewage treatment plant in Izabelin “Mokre Łąki”, Poland). LB medium was used for phage amplification (LB broth composition: 10.0 g L^−1^ of peptone (BLT, Poland), 10.0 g L^−1^ of sodium chloride (Chempur, Poland), and 5.0 g L^−1^ of yeast extract (BLT, Poland)). Phage culture medium containing 1 mL of phage lysate and 1 mL of an overnight bacterial host cultured on LB broth was incubated at 37 °C for 24 h. Afterward, the culture was centrifuged at 8000× *g* for 8 min to separate bacteria from the proliferated bacteriophages. The supernatant was filtered using a syringe filter with a membrane pore diameter of 0.45 µm (Sartorius, Germany) [64].

### 2.2. Evaluation of Phage Activity

Determination of phage titer [PFU mL^−1^] was performed based on the double-layer plate method [65,66]. The kinetic growth of bacteria infected with specific bacteriophages was measured in ten replicates using a Bioscreen C Pro automated growth analyzer (Yo AB Ltd., Growth Curves, Helsinki, Finland). In total, 180 µL of an overnight bacterial culture refreshed in LB broth was added to the wells, followed by the addition of 20 µL of the appropriate phage suspension or LB medium (control cultures). Phage lysates were prepared so that the value of MOI (multiplicity of infections) coefficient in each well reached 1.0. The plate was placed in the Bioscreen C Pro for 24 h with an average stirring intensity of 15 s before measurement. Optical density measurement was performed every 30 min using a broadband filter (OD_400–600_). Effectiveness of bacterial growth inhibition resulting from phage application was assessed with regard to control culture that contained only bacteria. Based on the obtained curves illustrating the dependence of the change in optical density and the duration of the culture, the specific growth rate coefficient (µ) was calculated according to the following formula:µ = (ln OD_max_ − ln OD_min_)/Δt(1)
where ln OD_max_—the natural logarithm of the maximum value of the exponential growth of the culture; ln OD_min_—the natural logarithm of the minimal value exponential growth of the culture; and Δt—duration of the exponential growth of culture (h).

### 2.3. Multiplication of Phage Particles

Phage lysates were subjected to extended culturing (24, 48, and 72 h) to obtain a solution with an increased amount of viral particles and lytic enzymes. In total, 200 mL of LB medium was inoculated with 1 mL of bacterial host and 1 mL of the appropriate lysate. Incubation was performed at 37 °C for 24 h. Obtained suspension was centrifuged at 8000× *g* for 10 min to separate the bacterial cell’s pellet. The supernatant was filtered using a 0.45 µm pore size syringe filter. The purified phage lysate was diluted 1:1 with 2 × LB medium, inoculated with 1 mL of overnight bacterial culture and incubated for another 24 h. The operation was repeated until a 72-hour culture was obtained. After each day of cultivation, phage titers were determined using the two-layer plate method [65,66] taking into account the dilution of the lysate with the medium. The effect of the number of phage particles obtained after extended cultivation on the optical density of bacterial culture was determined using a Bioscreen C Pro automated growth analyzer.

### 2.4. Isolation of Phage Enzymes

In this study two methods of enzyme isolation were used: enzyme precipitation with cold acetone [67] and ultracentrifugation at 25,000 rpm for 40 min (ultracentrifuge Sorvall LYNX 6000, Thermo Fisher Scientific, Watertown, MA, USA). To isolate phage proteins using acetone, 15 mL of ice-cold acetone (Avantor Performance Materials Poland S.A., Poland) was slowly added to 10 mL of the lysate. The solution was gently stirred and left at 4 °C for 24 h to precipitate proteins. After incubation, the solution was centrifuged at 7000× *g* for 5 min at 4 °C. The supernatant was removed, and the formed precipitate was left for 2 h in the laminar chamber to evaporate the residual acetone. Subsequently, 10 mL of physiological saline was added to the obtained precipitate. In order to check the effectiveness of isolation of bacterial envelope-degrading enzymes, obtained protein solution was spot-applied to a freshly cultured bacteria (host) on a nutrient agar medium. Appearance of a brightening zone was an indicator of the antimicrobial properties of the suspension. Second method of phage enzyme isolation was ultracentrifugation, which was subjected to 10 mL of each lysate. Concentration of the protein in the solution obtained after each day of culture was determined by the Bradford method using Coomassie Brilliant Blue G-250 reagent (Pol-Aura, Poland) and checked for reduction of bacterial cell division using a Bioscreen C Pro growth analyzer.

### 2.5. Statistical Analysis

Measurement of the bacteriophage concentration (in PFU mL^−1^) in phage lysates after 24-, 48-, and 72-hour cultures, as well as concentration of phage proteins (in mg mL^−1^) were carried out in triplicate. Obtained data were statistically assessed using Statistica 13.3 software suite (TIBCO Software Inc., Palo Alto, CA, USA). Analysis of variance (ANOVA) followed by Tukey’s test post-hoc (*p* < 0.05) was performed for each examined parameter to define homogenous groups, which in the tables were marked with identical letters.

## 3. Results and Discussion

### 3.1. Characteristics of Phage Plaques

Figure 1 shows the plaques of phages specific for the tested bacteria strains of the genus *Serratia* and *Enterobacter*. The examined phages differed in the width and transparency of the plaques. Among the tested phages, the plaques and halo zone of the *Serratia* phage KKP 3708 had the largest diameter against *Serratia liquefaciens* KKP 3654 (Figure 1B). Lytic bacteriophages that were used produced a characteristic translucent border on the periphery of plaques—the halo zone. This zone is formed through the degradation of the bacterial envelope by phage enzymes (depolymerases) released into the environment, which in turn facilitates the adsorption, infection and disintegration of bacterial cells [68,69]. The large diameter of the plaque indicates the small size of the bacteriophage, which diffuses more easily through the agar and infects adjacent bacterial cells [70,71]. The size of the halo zone is affected by the amount of enzyme secreted and the number of bacteriophages in the plaque [70].

### 3.2. Lytic Activity of Phages against Bacterial Hosts

The lytic activity of phages was determined using a Bioscreen C Pro growth analyzer. For this purpose, growth curves (optical density as a function of bacterial cell number; data unpublished) were plotted for each strain. For each phage strain, the growth curve of phage-infected microorganisms was plotted against the control bacterial culture. Changes in the optical density of tested bacteria strains after infection with specific phages are shown in Figure 2. The curves show the beginning and the end of the logarithmic growth phase at MOI 1.0. Specific growth phase coefficients are shown in Table 1; lower coefficients for the phage-infected bacterial culture indicate significant suppression of cell divisions in the logarithmic growth phase. In each case, there was a gradual decrease in optical density due to the lysis of bacteria cells. In the phage-infected cultures, a shortening of the logarithmic growth phase was observed compared to the control cultures.

The logarithmic growth phase of *Enterobacter cloacae* KKP 3262 lasted 12 h 30 min from the beginning of incubation and the optical density increased by 0.331 during this time. Infection with *Enterobacter* phage KKP 3082 resulted in a shortening of the logarithmic phase, which lasted 1 h 30 min. Optical density after phage infection increased by 0.012 during this time. The value of the specific growth rate coefficient (µ) for the culture after phage infection (µ = 0.061) was higher compared to the control culture (µ = 0.047) due to the significantly longer duration of the logarithmic phase in the control culture.

After infection of *Serratia liquefaciens* KKP 3654 with *Serratia* phage KKP 3708, complete disruption of bacteria cell divisions was observed. No distinct logarithmic growth phase was observed in the phage-infected bacterial culture. The optical density of the phage-infected culture increased by 0.028 during the logarithmic growth phase. The specific growth coefficient for the control culture was 0.031 and for the infected culture 0.003, which confirms the reduction of bacteria cell divisions.

Regarding the control *Enterobacter cloacae* KKP 3684 culture, the logarithmic growth phase lasted for the entire period of the study (24 h). The logarithmic growth phase of the phage-infected bacterial culture lasted for 1 h 30 min, and the optical density increased slightly during this time. After 2 h 30 min, a decrease in optical density was observed due to the inhibition of bacterial cell division. After 12 h of cultivation, the optical density increased, which could be related to the activation of the bacterial resistance mechanism the bacteriophages used. A higher specific growth rate was observed for the phage-infected culture compared to the control, which might be related to the excessively short duration of the logarithmic phase in the infected culture.

The logarithmic growth phase for *Serratia marcescens* KKP 3687 control culture lasted 20 h, with an optical density increase of 0.606. After infection with the specific *Serratia* phage KKP 3709, the logarithmic growth phase lasted 3 h. The optical density during the logarithmic growth phase of the phage-infected culture increased by 0.026. After 3 h of culture, a decrease in optical density was observed, probably due to the lysis of the bacterial cell. After 16 h of culturing, the optical density started to increase again, which could be due to the acquisition of defense mechanisms against phage infection [72]. A lower value of the specific growth rate coefficient was observed for the phage-infected culture (µ = 0.028) compared to the control culture (µ = 0.047).

Changes in the onset of the logarithmic growth phase in phage-infected cultures were reported by Mahmound et al. [73]. The growth of *Salmonella* Kentucky after phage infection at MOI 1.0 was delayed by all bacteriophages tested compared to control cultures. Complete inhibition of bacteria growth was observed after 24 h of incubation. A reduction of 2.65 log after 2 h of incubation of *Escherichia coli* O157:H7 infected with a specific phage was reported by Mozaffari et al. [74]. This downward trend continued for four hours after incubation and the difference in bacterial concentration between bacterial and bacteria–phage cultures was 3.8 log. After 4 h, an increased optical density was observed in this research [74], although its rate was much lower in phage-infected bacteria culture. In an experiment by Gientka et al. [11], different methods were used to apply the phage suspension to sprouts. The spraying method resulted in a reduction of the total number of bacteria to 1.5 log CFU g^−1^ after 48 h of incubation, while the application of absorption pads soaked in a phage cocktail reduced the total bacteria count to about 0.27–0.79 log CFU g^−1^.

### 3.3. Phage Concentration after Extended Propagation

Effective reproduction of phage virions takes place in an environment that is optimal for both bacteria and phages, and in the presence of living bacterial cells in the logarithmic growth phase [75,76]. In phage therapy, the concentration of phages in the preparation is limited by the size of bacteriophages because a too-viscous solution can be obtained [77]. The obtained phage titer results after each day of propagation are shown in Table 2. The removal of dead bacterial cells and the addition of the fresh culture and medium created optimal conditions for virion regeneration and enzyme release. A statistically significant constant increase in the concentration of phage particles in the solution was observed. The multiplication of the *Enterobacter* phage KKP 3262 lysate against *Enterobacter cloacae* KKP 3082 resulted in a steady increase in phage concentration by ca. 0.3 log per 24 h and significant differences (*p <* 0.05) between each day of culture. The highest increase in phage concentration was obtained for the *Serratia* phage KKP 3708 against *Serratia liquefaciens* KKP 3654. The cultivation of the control lysate for 24 h caused an increase in phage titer by 2 log. Conducting subsequent cultures increased phage activity by 0.2 log per day. The final phage lysate showed a concentration increase by above 2 log compared to the control lysate. *Serratia* phage KKP 3709 against *Serratia marcescens* KKP 3687 had the weakest ability to multiply. After the first day of cultivation, no significant difference (*p* < 0.05) in phage particle concentration was observed. Cultivation for 72 h resulted in an increase in titer by almost half a logarithm compared to the control lysate. Cultivation of *Enterobacter* phage KKP 3711 against *Enterobacter cloacae* KKP 3684 on each successive day resulted in an increase in the phage titer in the lysate by an average of 0.2 log. A 72-hour culture yielded a lysate with an increased bacteriophage count by ca. half a logarithmic order compared to the control culture.

To optimize large-scale phage production, numerous studies were conducted on continuous multiplication. The addition of bacterial cultures before the stationary phase results in better viral replication [78,79]. Studies showed low bacteriophage multiplication efficiency in long-term continuous phage production in a bioreactor due to the prolonged simultaneous presence of phage and bacteria, resulting in spontaneous mutations of both [80,81]. For optimal continuous phage production, environmental conditions (such as temperature, pH, and culture medium) affecting bacterial growth and therefore indirectly affecting phage formation should also be considered. The composition of the culture medium also significantly affects the physiological state of bacteria, including cellular composition and metabolism, and thus impacts phage formation [82,83]. In the case of extreme substrate limitation, phage formation can be completely stopped despite infection, as is the case of hibernation mode [84].

### 3.4. Measurement of Phage Enzymes Concentration

Depolymerase activity is commonly identified by the ever-expanding halo zone that surrounds the phage bald spot [85]. Studies show the effective eradication of bacteria in vitro regardless of the growth phase of bacterial cells through the action of phage endolysins [85]. In this study, obtained phage suspensions after 24-, 48- and 72-hour cultures were subjected to phage enzyme concentration using cold acetone and the ultracentrifugation method. The culture of obtained protein suspensions led to the formation of lysis zones of bacterial cells on the agarose medium.

The results of increasing protein concentration in successive hours of bacterial cultures are shown in Table 3. The highest protein concentration was obtained after 72 h culture of *Serratia* phage KKP 3708 against *Serratia liquefaciens* KKP 3654 after isolation with acetone. Protein concentration after the last day of cultivation increased by 0.063 mg mL^−1^ compared to the cultivation performed for 24 h. The use of the ultracentrifugation method after each day of propagation showed significant differences (*p* < 0.05) in protein concentration. It is noteworthy that among the studied bacteriophages, *Serratia* phage KKP 3708 had the largest halo zone diameter, indicating high production of extracellular enzyme proteins. Among the tested phages, an increase in the concentration of almost 30% after a 72 h culture was shown by *Serratia* phage KKP 3709 against *Serratia liquefaciens* KKP 3687 after using ultracentrifugation as a method of protein isolation. The applied methods of enzyme isolation led to significantly different outcomes (*p* < 0.05). It is noteworthy that the phage studied had an effective increase in the proliferation rate of phage particles during prolonged culture. Protein concentration after 72 h culture increased by 0.028 mg mL^−1^ compared to the 24 h culture. Lysates of this bacteriophage were characterized by faintly visible halo zones. The concentration of proteins isolated by ultracentrifugation for the *Enterobacter* phage KKP 3262 showed significant differences (*p* < 0.05) between the days of propagation. No significant differences were observed between 48 h culture and 72 h culture with the enzymes isolated using acetone. The highest protein concentration for *Enterobacter* phage KKP 3711 was obtained after 48 h culture using acetone precipitation.

The conducted study confirms the release of phage enzymes into the environment during the lysis of the host bacterial cells. The diameters of the halo zones were correlated with the results of the obtained enzyme concentration. Absorbance values for culture after using ultracentrifugation as a method of protein isolation were slightly higher compared to absorbance values obtained using acetone, which may be related to the presence of other absorbents (components of culture medium) in addition to the labeled component. The use of the described methods makes it possible to obtain phage enzymes and reduce the number of phage particles in the lysate. In turn, chromatographic purification technology allows the complete separation of enzymes from phage virions. In the study conducted by Jurczak-Kurek et al. [86], most of the examined bacteriophages were inactivated because of the treatment with 90% acetone. The action of acetone on phage enzymes consists of the weakening of hydrophobic interactions and the consequent denaturation of proteins, leading to the formation of a protein precipitate [86]. A study by Diez-Martines et al. [87] confirms the effectiveness of using a small number of phage enzymes. Incubation of the Cpl-1 lysine at a concentration of 5 ng µL^−1^ with *S. pneumoniae* strain R6 for 60 min at 37 °C resulted in the eradication of the bacterial culture. Van de Kamp described a 50% reduction in optical density after adding 64 ng µL^−1^ of lysine 23TH_48 against *S. pneumoniae* R6 for 15 min at 37 °C. The addition of 1–1.5 ng µL^−1^ of lysine 23TH_48 to *S. pneumoniae* DSM 24048 led to the reduction of bacteria count by 4 log CFU mL^−1^ in 1 h.

### 3.5. Changes in the Optical Density of Bacterial Cultures after Phage Infection

The effectiveness of the obtained phage and enzyme suspensions in bacterial reduction was examined in vitro. For this purpose, the Bioscreen C Pro growth analyzer was used and the obtained results are shown in Table 4.

Changes in the optical density of the tested strain cultures are shown in Figure 3. The logarithmic growth phase of the *Enterobacter cloacae* KKP 3082 control culture lasted 6 h 30 min, and optical density increased by 0.271 during this time. The logarithmic growth phase of bacterial culture after infection with *Enterobacter* phage KKP 3262 suspensions after 24-, 48- and 72-hour cultures proceeded in a similar way and lasted only 30 min. After 30 min of incubation, a decrease in optical density was observed, which indicated the inhibition of bacterial cell division. Determining specific growth rate coefficients for each phage-infected culture (µ = 0.007) and control culture (µ = 0.055) confirmed the reduction in the bacterial division. This result can be explained by the better effect of phage enzymes on bacteria in the logarithmic growth phase, because in the stationary growth phase LPS (lipopolysaccharide) modifications may occur, which reduces the antibacterial effect [88]. There were no differences in the growth of *Serratia liquefaciens* KKP 3654 after infection with *Serratia* phages KKP 3708 after 24-, 48- and 72-hour multiplying. The logarithmic growth phases after infection and in the control bacteria culture were 1 h 30 min and 16 h 30 min, respectively. After the logarithmic growth phase in the culture with a phage, a decrease in optical density was observed, probably due to the synergistic effect of enzymes and bacteriophages on bacterial cells. Determining specific growth rate coefficients for control cultures (µ = 0.06) and cultures after 72-hour infection (µ = 0.048) indicate a reduction in the number of bacterial cells. The logarithmic growth phase of the *Enterobacter cloacae* KKP 3684 control culture lasted 7 h, while optical density increased by 0.439. Due to infection of bacterial culture with a 72-hour *Enterobacter* phage KKP 3711 multiplied culture, the logarithmic growth phase lasted for 1 h and optical density increased by 0.011 during this time. The determined coefficients (µ = 0.085 for control and µ = 0.02 for bacterial culture after 72 h infection) confirm the reduction in bacteria count. Regarding the infection with the *Serratia* phage KKP 3709 after 72 h propagation, the logarithmic growth phase lasted 4 h 30 min—one hour shorter than the *Serratia marcescens* KKP 3687 control culture. The optical density increased in the course of this phase by 0.106 and 0.475, respectively.

The obtained results indicate that the use of lysates derived from extended culture can significantly shorten the logarithmic phase against control culture, as well as decrease specific growth rates. The observed phenomenon indicates an increased concentration of phage enzymes in the environment, which facilitated the occurrence of effective phage infection. Yazdi et al. [89] reported that the application of higher phage concentrations resulted in a faster reduction of the bacterial cells, which could be due to the increased rate of attachment of higher bacteriophage titers.

Rainovic et al. [90] studied the dependence of the efficiency of phage reconstitution and the concentration of bacteria and bacteriophages. The results indicate that high bacteria concentrations (10^8^ CFU mL^−1^) combined with low phage concentrations (10^4^, 10^3^, 10^2^, and 10^1^ PFU mL^−1^) result in unrestricted bacterial growth, which in most cases cannot be distinguished from the growth kinetics of the control. Cultures with high cell concentrations are not sensitive to low phage counts because the culture reaches the stationary phase before the phage has had time to multiply sufficiently to cause detectable lysis. Low bacterial concentrations (10^6^ CFU mL^−1^) combined with high phage concentrations (10^6^, 10^7^, and 10^8^ PFU mL^−1^) do not show detectable growth of bacteria, as the entire culture is lysed before the optical density begins to increase.

### 3.6. Influence of Phage Enzymes on the Growth Kinetics of Bacterial Host Strains

In order to determine the effectiveness of inhibiting bacterial growth by the enzymes isolated from 24-, 48-, and 72-hour cultures, the optical density of the cultures was measured. The obtained results of the average optical density and the determined specific growth rate coefficients are shown in Table 5, Figure 4, and Figure 5. In all cases, the enzymes isolated from the 72 h culture exhibited the best inhibitory effect on bacterial growth, therefore the following description concerns the effect of enzymes isolated from the 72 h culture. Phage enzymes isolated through ultracentrifugation reduced *Enterobacter cloacae* KKP 3082 growth the most efficiently. The specific growth rate coefficients of bacterial cultures after the application of enzymes isolated with the use of acetone (72 h propagation; µ = 0.077) were higher compared to the rates of bacterial cultures with the addition of enzymes isolated by the ultracentrifugation method (72 h propagation; µ = 0.021). The method of enzyme isolation did not significantly affect bacterial growth curves. The logarithmic growth phase for the control lasted 6 h and optical density increased by 0.254. In bacterial cultures with the addition of the enzymes isolated by acetone and ultracentrifugation, the logarithmic growth phase lasted 1 h 30 min and the optical density increased by an average of 0.027 during this time.

The logarithmic growth phase of the *Serratia liquefaciens* KKP 3654 control culture lasted 9 h and the optical density increased by 0.495 during this time. A reduction of optical density in the bacterial culture with enzymes after the log phase was observed. The addition of enzymes (72 h propagation) isolated by ultracentrifugation resulted in a prolonged log phase (2 h 30 min) compared to the effect of enzymes isolated by acetone (1 h 30 min), and optical density during the logarithmic growth increased by 0.114 and 0.049, respectively. In most cases, coefficients of the specific growth rate indicated an inhibitory effect on bacteria growth, and coefficients of the method with acetone were lower compared to the ultracentrifugation method. Regarding the *Enterobacter cloacae* KKP 3684 control culture, the logarithmic growth phase lasted 6 h and optical density increased by 0.322 during this time. After the addition of phage enzymes (72 h propagation), the logarithmic growth phase lasted 1 h 30 min for enzymes isolated by acetone and 2 h for enzymes isolated through ultracentrifugation, and the optical density increased by 0.007 and 0.044, respectively, during this time. The specific growth rate coefficient for culture after adding enzymes isolated with acetone was high in comparison to the control. This phenomenon can be explained by the excessively short duration of logarithmic growth of bacteria cultured with enzymes. Considering the addition of enzymes isolated through ultracentrifugation, specific growth rate coefficients showed a significant inhibitory effect on bacteria growth.

The logarithmic growth phase of the *Serratia marcescens* KKP 3687 control culture lasted 6 h. The optical density increased by 0.397 during this time. The determined specific growth rate coefficients (for control culture: µ = 0.088, for cultures with added phage proteins isolated through ultracentrifugation and with acetone after 72-hour propagation: µ = 0.036) testify to the reduction in the growth of bacteria strain due to the application of isolated enzymes.

Phage enzymes are promising antimicrobial agents with potential applications in the food sector [17]. In the research conducted by Zhang et al. [91], LysSTG2 endolysin was used to control *Pseudomonas aeruginosa* and *P. putida*, which showed high sensitivity in bottled water, chicken breast, and milk. In bottled water, LysSTG2 in combination with EDTA reduced the number of tested bacteria by 2.2 log and below the detection limit, respectively. Contamination of pasteurized milk with the examined bacteria did not result in a significant decrease in the number of viable bacteria cells. The addition of LysSTG2 alone at a concentration of 1 mg mL^−1^ reduced the number of living bacteria by 1.0 log in salmon samples and 0.6 log in chicken [91]. Some endolysins show low levels of expression and insolubility [92], and their antibacterial efficiency depends on food components (fat, proteins, carbohydrates, vitamins) and biochemical factors (pH, temperature) [93]. In the study conducted by Yan et al. [94], sodium chloride was found to have the ability to affect the binding capacity of endolysin. LysGH15 eradicated *Staphylococcus aureus* in whole and skimmed milk supplemented with NaCl. García et al. [95] also observed a strong synergistic effect in milk when LysH5 was combined with nisin. Various endolysins showed lytic activity against *S. aureus* in milk [95,96]. Regarding antimicrobial agents for plant products, several endolysins were suggested as potential candidates against *Listeria monocytogenes* in iceberg lettuce [40]. In addition to the direct antimicrobial effect in food matrices, the eradication of biofilm in food processing constitutes another important factor contributing to food safety assurance [97]. The synergistic effect of LysK endolysin and DA7 depolymerase on *Staphylococcus* biofilms (DP) was also proven. Even low concentrations of nano mixtures of these enzymes were effective in removing biofilms from glass and polystyrene surfaces [97]. A synergistic effect of bacteriophage depolymerase and chlorine dioxide application for eradication of *Klebsiella* sp. biofilm was observed. After 4 h of biofilm exposure to DP phage, an 80% reduction in bacterial cell number was achieved. It was also noted that treatment of *Klebsiella* biofilm with DP for 4 h, followed by a 30-min treatment with chlorine dioxide, resulted in the elimination of 92% of the biofilm-forming bacteria [98]. Currently, there are many studies concerning the simultaneous application of phage enzymes and high hydrostatic pressure (HHP). Van Nassau et al. [99] reported a 5.5 log reduction in the number of *L. monocytogenes* after the application of *L. monocytogenes*-specific endolysins Ply825 (0.16 µg mL^−1^) and HHP (300 MPa, 1 min, 30 °C). The synergistic effect of HHP and endolysin also enabled the effective applications of lower HHP parameters (200 MPa, 2 min, 30 °C) [99]. Endolysins that exhibit activity against food-borne pathogens were frequently studied in food matrices. In addition, several studies have confirmed the safety of endolysins with respect to human health [100,101]. In conclusion, phage enzymes are promising biocontrol agents that have the potential to be used in food as well as in food-producing facilities.

In our study, in most cases, the addition of enzymes isolated after 72 h of phage propagation showed better restriction of bacterial cell divisions, which is correlated with a higher concentration of phage proteins in the solutions. In vitro studies suggest that the use of isolated phage enzymes can effectively reduce saprophytic bacteria, which represent a food contaminant. In order to better assess the potential of phage enzymes, their performance should be evaluated in different ranges of pH, temperature, and ionic strength, using different environmental conditions and cells in different physiological stages. The obtained results show that the combination of phages and their enzymes can be an effective method to reduce the bacteria count.

## 4. Conclusions

Many researchers focus on the application of phages and their enzymes to combat pathogenic bacteria, while in minimally processed food products the problem of contamination with saprophytic bacteria remains neglected. This is probably due to the lack of strict microbiological quality guidelines in the context of monitoring the total number of microorganisms in these products. Therefore, in our research, we drew attention to the problem of RTE food product contamination with saprophytic bacteria and indicated the effectiveness of an application of phages and their enzymes to reduce saprophytic bacterial growth in vitro, which consequently can translate into improved food quality and ensure consumer health safety. An extended time of phage particle propagation in bacterial cultures enables an increase in the number of phages and their enzymes. Introducing bacterial cells at 24 h intervals during phage particle reconstruction can lead to a 100-fold increase in the number of virions in the lysates. An increased number of phages in the lysates resulted in an increased concentration of proteins (ca. 20%) that exhibit lytic activity. The combination of phages and their enzymes exhibits a synergistic effect and makes the process of elimination of saprophytic bacteria more effective. Therefore, *Enterobacter* phage KKP 3082, *Enterobacter* phage KKP 3684, *Serratia* phage KKP 3654, and *Serratia* phage KKP 3687 represent promising novel biocontrol agents and may be applied in the future to control the dominant saprophytic bacteria in RTE plant-based food products. Both phages and their enzymes can promote a reduction in antibiotic use, which is a matter of great importance given the alarming rise in antibiotic resistance. In addition to their potential ability to specifically control both saprophytic and pathogenic bacteria, their use does not generate negative environmental impacts in contrast to antibiotics. Phages and phage lysins have opened up a new step in food preservatives and food safety measures for use throughout the food chain.

## Figures and Tables

**Figure 1 viruses-15-00009-f001:**
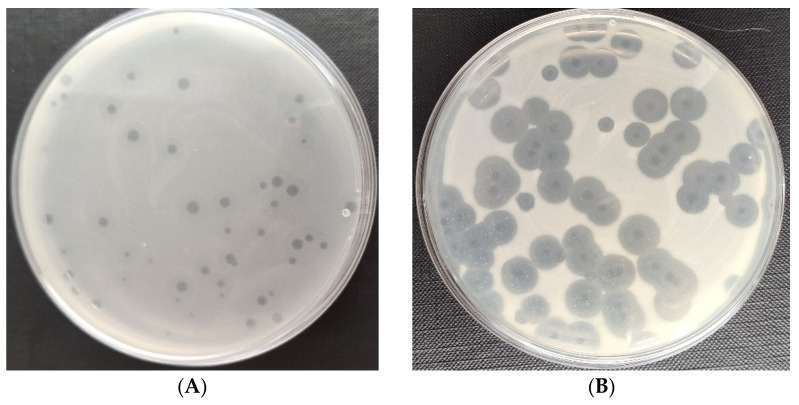
Plaques of specific phages against the bacterial hosts: (**A**) *Enterobacter* phage KKP 3262; (**B**) *Serratia* phage KKP 3708; (**C**) *Enterobacter* phage KKP 3711; (**D**) *Serratia* phage KKP 3709.

**Figure 2 viruses-15-00009-f002:**
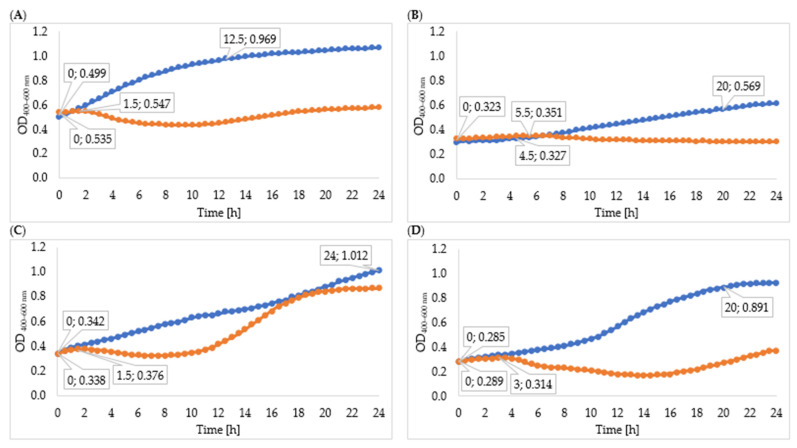
Growth curves of tested bacteria: (**A**)—*Enterobacter cloacae* KKP 3082; (**B**)—*Serratia liquefaciens* KKP 3654; (**C**)—*Enterobacter cloacae* KKP 3684; (**D**)—*Serratia marcescens* KKP 3687. The figure shows the beginning and the end of the logarithmic growth phase (MOI 1.0), *n* = 10. Blue line—control culture, orange line—phage-infected bacterial culture.

**Figure 3 viruses-15-00009-f003:**
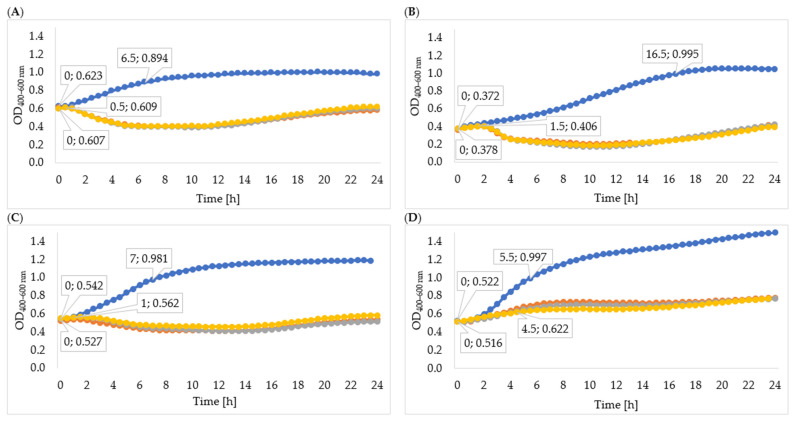
Growth curves of tested bacteria: (**A**)—*Enterobacter cloacae* KKP 3082; (**B**)—*Serratia liquefaciens* KKP 3654; (**C**)—*Enterobacter cloacae* KKP 3684; (**D**)—*Serratia marcescens* KKP 3687. The figure shows the beginning and the end of the logarithmic growth phase for control and tested bacterial strains after addition of specific phages multiplied for 72 h (MOI 1.0), *n* = 10. Blue line—control culture, orange line—culture infected with phages propagated for 24 h, grey line—culture infected with phages propagated for 48 h, yellow line—culture infected with phages propagated for 72 h.

**Figure 4 viruses-15-00009-f004:**
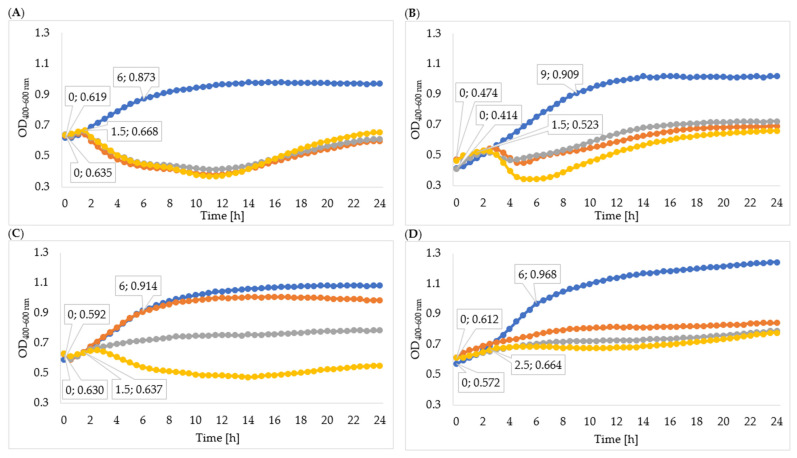
Growth curves of tested bacteria cultured with enzymes isolated with acetone: (**A**)—*Enterobacter cloacae* KKP 3082; (**B**)—*Serratia liquefaciens* KKP 3654; (**C**)—*Enterobacter cloacae* KKP 3684; (**D**)—*Serratia marcescens* KKP 3687. The beginning and the end of logarithmic growth phases for control and tested bacteria strains cultured with enzymes isolated after 72-hour propagation are marked in the figure (*n* = 10). Blue line—control culture, orange line—culture infected with phages propagated for 24 h, grey line—culture infected with phages propagated for 48 h, yellow line—culture infected with phages propagated for 72 h.

**Figure 5 viruses-15-00009-f005:**
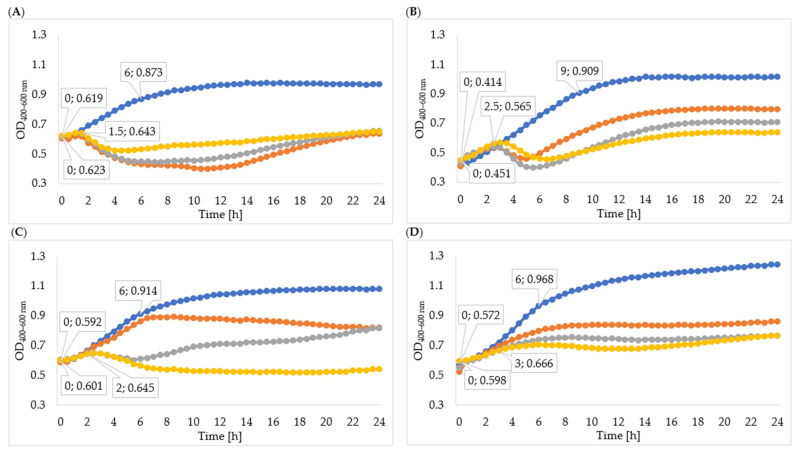
Growth curves of tested bacteria cultured with enzymes isolated through ultracentrifugation: (**A**)—*Enterobacter cloacae* KKP 3082; (**B**)—*Serratia liquefaciens* KKP 3654; (**C**)—*Enterobacter cloacae* KKP 3684; (**D**)—*Serratia marcescens* KKP 3687. The beginning and the end of logarithmic growth phases for control and tested bacteria strains cultured with enzymes isolated after 72-hour propagation are marked in the figure (*n* = 10). Blue line—control culture, orange line—culture infected with phages propagated for 24 h, grey line—culture infected with phages propagated for 48 h, yellow line—culture infected with phages propagated for 72 h.

**Table 1 viruses-15-00009-t001:** Values of specific growth rate coefficients of the tested bacteria cultures after addition of specific phages (*n* = 10).

Bacterial Host Strain	Phage Strain	Control Culture		Phage-Infected Bacterial Culture	
		ΔOD	µ [h^−1^]	ΔOD	µ [h^−1^]
*Enterobacter cloacae* KKP 3082	*Enterobacter* phage KKP 3262	0.470	0.047	0.012	0.061
*Serratia liquefaciens* KKP 3654	*Serratia* phage KKP 3708	0.242	0.031	0.004	0.003
*Enterobacter cloacae* KKP 3684	*Enterobacter* phage KKP 3711	0.675	0.046	0.038	0.071
*Serratia marcescens* KKP 3687	*Serratia* phage KKP 3709	0.606	0.047	0.025	0.028

**Table 2 viruses-15-00009-t002:** Bacteriophage concentration in phage lysates after consecutive hours of culture.

Phage Strain	Control Culture	24 h Culture	48 h Culture	72 h Culture
	[log PFU mL^−1^]			
*Enterobacter* phage KKP 3262	8.20 ± 0.03 ^d^	8.57 ± 0.02 ^c^	8.78 ± 0.04 ^b^	9.00 ± 0.02 ^a^
*Serratia* phage KKP 3709	9.99 ± 0.06 ^c^	9.87 ± 0.06 ^d^	10.25 ± 0.02 ^b^	10.38 ± 0.03 ^a^
*Serratia* phage KKP 3708	7.10 ± 0.03 ^d^	9.00 ± 0.07 ^c^	9.20 ± 0.06 ^b^	9.41 ± 0.07 ^a^
*Enterobacter* phage KKP 3711	9.74 ± 0.05 ^d^	10.00 ± 0.03 ^c^	10.24 ± 0.04 ^b^	10.38 ± 0.04 ^a^

Lowercase letters in the same rows are significantly different at *p* < 0.05 (*n* = 3).

**Table 3 viruses-15-00009-t003:** Concentration of phage proteins in solution after isolation.

Phage Strain	Enzyme Isolation Method	Concentration [mg mL^−1^]		
		24 h Culture	48 h Culture	72 h Culture
*Enterobacter* phage KKP 3262	Precipitation with acetone	0.275 ± 0.008 ^b^	0.300 ± 0.003 ^a^	0.303 ± 0.003 ^a^
	Ultracentrifugation	0.289± 0.002 ^c^	0.304 ± 0.003 ^b^	0.321 ± 0.001 ^a^
*Serratia* phage KKP 3709	Precipitation with acetone	0.273 ± 0.004 ^b^	0.276 ± 0.002 ^b^	0.319 ± 0.001 ^a^
	Ultracentrifugation	0.289 ± 0.002 ^c^	0.324 ± 0.003 ^b^	0.352 ± 0.002 ^a^
*Serratia* phage KKP 3708	Precipitation with acetone	0.297 ± 0.005 ^b^	0.301 ± 0.003 ^ab^	0.360 ± 0.000 ^a^
	Ultracentrifugation	0.292 ± 0.002 ^b^	0.287 ± 0.002 ^c^	0.304 ± 0.002 ^a^
*Enterobacter* phage KKP 3711	Precipitation with acetone	0.299 ± 0.002 ^c^	0.328 ± 0.002 ^a^	0.311 ± 0.003 ^b^
	Ultracentrifugation	0.291 ± 0.001 ^b^	0.313 ± 0.003 ^a^	0.317 ± 0.001 ^a^

Lowercase letters in the same rows are significantly different at *p* < 0.05 (*n* = 3).

**Table 4 viruses-15-00009-t004:** Values of specific growth rate coefficient of tested bacterial cultures after addition of phages multiplied for 72 h (*n* = 10).

Bacterial Host Strain		Control Bacterial Culture		Phage-Infected Bacterial Culture	
		ΔOD	µ [h^−1^]	ΔOD	µ [h^−1^]
*Enterobacter cloacae* KKP 3082	Multiply 24 h	0.271	0.055	0.004	0.013
	Multiply 48 h			0.004	0.013
	Multiply 72 h			0.002	0.007
*Serratia liquefaciens* KKP 3654	Multiply 24 h	0.623	0.060	0.033	0.080
	Multiply 48 h			0.033	0.056
	Multiply 72 h			0.028	0.048
*Enterobacter cloacae* KKP 3684	Multiply 24 h	0.439	0.085	0.012	0.023
	Multiply 48 h			0.014	0.025
	Multiply 72 h			0.035	0.020
*Serratia marcescens* KKP 3687	Multiply 24 h	0.475	0.118	0.199	0.050
	Multiply 48 h			0.150	0.042
	Multiply 72 h			0.106	0.035

**Table 5 viruses-15-00009-t005:** Values of specific growth rate coefficient of tested bacterial cultures after addition of enzymes after 72 h propagation (*n* = 10).

Bacterial Host Strain		Control Bacterial Culture		Enzymes Isolated with Acetone		Enzymes Isolated through Ultracentrifugation	
		ΔOD	µ [h^−1^]	ΔOD	µ [h^−1^]	ΔOD	µ [h^−1^]
*Enterobacter cloacae* KKP 3082	Multiply 24 h	0.254	0.057	0.012	0.038	0.013	0.014
	Multiply 48 h			0.014	0.043	0.025	0.027
	Multiply 72 h			0.033	0.077	0.020	0.021
*Serratia liquefaciens* KKP 3654	Multiply 24 h	0.495	0.087	0.072	0.057	0.122	0.087
	Multiply 48 h			0.111	0.120	0.095	0.077
	Multiply 72 h			0.049	0.065	0.114	0.079
*Enterobacter cloacae* KKP 3684	Multiply 24 h	0.322	0.068	0.381	0.091	0.260	0.060
	Multiply 48 h			0.121	0.080	0.031	0.019
	Multiply 72 h			0.007	0.097	0.044	0.020
*Serratia marcescens* KKP 3687	Multiply 24 h	0.397	0.088	0.141	0.038	0.263	0.074
	Multiply 48 h			0.063	0.033	0.158	0.056
	Multiply 72 h			0.052	0.037	0.068	0.036

## Data Availability

Not applicable.

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
