# Peer review of "Application of Lytic Bacteriophages and Their Enzymes to Reduce Saprophytic Bacteria Isolated from Minimally Processed Plant-Based Food Products—In Vitro Studies"

_viruses, 2022, doi:10.3390/v15010009_

Round 1

Reviewer 1 Report

The topic of using phage lytic enzymes is gaining an increasing interest nowadays, therefore the studies on various phage produced lysins or holins are of high importance. The results presented by the authors are not exactly of high importance, they are rather preliminary. However, they show a potential to be investigated in more detail in the future and provide more data on the activity and specyfity of this enzymes.

Few comments:

1. have the phages been characterized and published previousely? If so, please provide the reference in the methods section as well as i.e. GenBank numbers for the phages, so it will be easier to find. if they are novel, please state it in the text.

2.  Please change the layout of the graphs in Fig 2 -5. Show the x and y axis, remove the lines and mark the graphs as A,B etc. either on top or in the top left corner so it will be more coherent with other published data and easier to read. Consider use of 10log scale.

3. Also the description of the graphs is misleading. The rewiever understands that they show the activity of enzymes from phage lyzates grown for different period of time while the description suggest it's the culture infected with phages obtained from the culture grown for a certain amout of time.

4. Where the enzymes specyfic? The problems with lytic enzymes is that they have a broader spectrum that the phage itself. If such tests were done, please state it in the text and provide data. If not, the rewiever suggests this should be done.

Author Response

Review Report (Reviewer 1)

The topic of using phage lytic enzymes is gaining an increasing interest nowadays, therefore the studies on various phage produced lysins or holins are of high importance. The results presented by the authors are not exactly of high importance, they are rather preliminary. However, they show a potential to be investigated in more detail in the future and provide more data on the activity and specyfity of this enzymes.

Dear Reviewer,

Thank you for reading our manuscript carefully and for your critical review. We have addressed all your comments below. All changes introduced to the manuscript are marked in yellow. In this document, responses to your suggestions are also marked in yellow.

Few comments:

  1. have the phages been characterized and published previousely? If so, please provide the reference in the methods section as well as i.e. GenBank numbers for the phages, so it will be easier to find. if they are novel, please state it in the text.

It has been corrected in Materials and Methods section (lines 141-145). One of the phages (Enterobacter phage KKP 3262) was characterized in the article conducted by Wójcicki et al. (2021):

Wójcicki, M.; Åšrednicka, P.; BÅ‚ażejak, S.; Gientka, I.; Kowalczyk, M.; Emanowicz, P.; Åšwider, O.; SokoÅ‚owska, B.; Juszczuk-Kubiak, E. Characterization and Genome Study of Novel Lytic Bacteriophages against Prevailing Saprophytic Bacterial Microflora of Minimally Processed Plant-Based Food Products. Int. J. Mol. Sci. 2021, 22, 12460.

Other three bacteriophages are novel and will be characterized in our future study.

  1. Please change the layout of the graphs in Fig 2 -5. Show the x and y axis, remove the lines and mark the graphs as A,B etc. either on top or in the top left corner so it will be more coherent with other published data and easier to read. Consider use of 10log scale.

Thank you for your comment. It has been corrected according to your suggestions.

  1. Also the description of the graphs is misleading. The rewiever understands that they show the activity of enzymes from phage lyzates grown for different period of time while the description suggest it's the culture infected with phages obtained from the culture grown for a certain amout of time.

Thank you for your comment. It has been corrected in our manuscript.

  1. Where the enzymes specyfic? The problems with lytic enzymes is that they have a broader spectrum that the phage itself. If such tests were done, please state it in the text and provide data. If not, the rewiever suggests this should be done.

Yes, we agree with Reviewer's opinion, but this aspect of research is currently being conducted by us and will be included in our next article.

We have tried our best to improve the Manuscript. We hope that the introduced changes will allow our article to be published in Viruses.

Sincerely,

Dziyana Shymialevich

MichaÅ‚ Wójcicki

Responses to the Reviewer 1 comments are included in the attachment.

Reviewer 2 Report

This manuscript reports on the isolation of phages and their enzymes to apply them on in vitro assays for elimination of saprophytic bacteria isolated from minimally processed food.

- Missing discussion / sections were not exhaustively discussed in the following sections:

Section 3.2. “Lytic Activity of Phages Against Bacterial Hosts”

Section 3.3 “Phage Concentration After Extended Propagation”

Section 3.4 “Measurement of Phage Enzymes Concentration”

Section 3.5. “Changes in the Optical Density of Bacterial Cultures After Phage Infection”

The results obtained in this work should be discussed in relation to the results found by other authors.

- The manuscript does not highlight the novelty of the work.

Author Response

Review Report (Reviewer 2)

This manuscript reports on the isolation of phages and their enzymes to apply them on in vitro assays for elimination of saprophytic bacteria isolated from minimally processed food.

Dear Reviewer,

Thank you for reading our manuscript carefully and for your critical review. We have addressed all your comments below. All changes introduced to the manuscript are marked in yellow. In this document, responses to your suggestions are also marked in yellow.

- Missing discussion / sections were not exhaustively discussed in the following sections:

Section 3.2. “Lytic Activity of Phages Against Bacterial Hosts”

The discussion in this section has been expanded (lines 303-307).

Section 3.3 “Phage Concentration After Extended Propagation”

The discussion in this section has been expanded (lines 337-348).

Section 3.4 “Measurement of Phage Enzymes Concentration”

The discussion in this section has been expanded (lines 410-416).

Section 3.5. “Changes in the Optical Density of Bacterial Cultures After Phage Infection”

The discussion in this section has been expanded (lines 484-493).

The results obtained in this work should be discussed in relation to the results found by other authors.

Thank you for your valuable comment. According to your suggestion, additional discussion in relation to available literature data has been introduced to our manuscript.

- The manuscript does not highlight the novelty of the work.

Thank you for your valuable comment. According to your suggestion, the novelty of our work has been underlined (lines 596-604 and 610-619).

We have tried our best to improve the Manuscript. We hope that the introduced changes will allow our article to be published in Viruses.

Sincerely,

Dziyana Shymialevich

MichaÅ‚ Wójcicki

Responses to the Reviewer 2 comments are included in the attachment.
